# Electrical Monitoring as a Novel Route to Understanding the Aging Mechanisms of Carbon Nanotube-Doped Adhesive Film Joints

**Xoan F. Sánchez-Romate** [1,2,*], **Alberto Jiménez-Suárez** [2], **María Sánchez** [2], **Silvia G. Prolongo** [2], **Alfredo Güemes** [1] and **Alejandro Ureña** [2]

1   Department of Aerospace Materials and Processes, Escuela Técnica Superior de Ingeniería Aeronáutica y del Espacio, Universidad Politécnica de Madrid, Plaza del Cardenal Cisneros 3, 28040 Madrid, Spain; alfredo.guemes@upm.es

2   Materials Science and Engineering Area, Escuela Superior de Ciencias Experimentales y Tecnología, Universidad Rey Juan Carlos, Calle Tulipán s/n, 28933 Móstoles (Madrid), Spain; alberto.jimenez.suarez@urjc.es (A.J.-S.); maria.sanchez@urjc.es (M.S.); silvia.gonzalez@urjc.es (S.G.P.); alejandro.urena@urjc.es (A.U.)

*   Correspondence: xoan.fernandez.sanchezromate@urjc.es; Tel.: +34914884771



**Featured Application: Structural Health Monitoring of novel carbon nanotube doped adhesive joints under aging conditions.**

**Abstract:** Carbon fiber-reinforced plastic bonded joints with novel carbon nanotube (CNT) adhesive films were manufactured and tested under different aging conditions by varying the surfactant content added to enhance CNT dispersion. Single lap shear (SLS) tests were conducted in their initial state and after 1 and 2 months immersed in distilled water at 60 °C. In addition, their electrical response was measured in terms of the electrical resistance change through thickness. The lap shear strength showed an initial decrease due to plasticization of weak hydrogen bonds, and then a partial recovery due to secondary crosslinking. This plasticization effect was confirmed by differential scanning calorimetry analysis with a decrease in the glass transition temperature. The electrical response varied with aging conditions, showing a higher plasticity region in the 1-month SLS joints, and a sharper increase in the case of the non-aged and 2-month-aged samples; these changes were more prevalent with increasing surfactant content. By adjusting the measured electrical data to simple theoretical calculations, it was possible to establish the first estimation of damage accumulation, which was higher in the case of non-aged and 2-month-aged samples, due to the presence of more prevalent brittle mechanisms for the CNT-doped joints.

**Keywords:** carbon nanotubes; aging; structural health monitoring; water uptake; adhesive film; surfactant

## 1. Introduction

The increasing requirements of industry in terms of structural components make the development of novel materials necessary. In this context, composite structures present many advantages over conventional metallic alloys due to their exceptional specific properties that lead to energy efficiency and weight savings.

Therefore, the assembly of several composite parts is a challenging subject as the complexity of these structures is continuously increasing. For these purposes, bonded joints have some advantages over bolted ones as they are lightweight and avoid stress concentrations around the holes [1].

However, the inspection of adhesive joints sometimes is not always straightforward, since it involves many complex techniques, such as fiber Bragg grating sensors or Lamb waves, which often do not give a complete overview of the quality of the bonded joint [2–4]. Therefore, it is necessary to develop novel procedures that do not involve complex data analysis techniques and are not detrimental to the physical properties of the joint.

In this regard, carbon nanotubes (CNTs) seem to be a very promising solution. Their exceptional properties [5–7] and the enhancement of the electrical conductivity that they induce when added to an insulator resin [8–11] makes them very useful for multifunctional applications [12,13]. In fact, their use in structural health monitoring (SHM) applications is now of interest because of their piezoresistive and tunneling properties that lead to high sensitivities [14–17].

The aim of this work is to exploit the superb physical properties of CNTs in developing novel multifunctional adhesives with an inherent self-sensing capability. To date, most research into reinforced bonded joints has been focused on paste adhesives [18–21]. They exhibit excellent sensing properties and are capable of properly monitoring strain and debonding [22–24]. These paste adhesives can be treated as nanoreinforced composites, with the CNT dispersion procedure representing a challenging subject that often involves complex and expensive techniques, such as three-roll milling [25–27]. Therefore, this work is focused on the effect that CNT addition has on adhesive films, which allows for better thickness control and is used for structural applications in the aircraft industry.

In previous studies, CNT reinforced adhesive films have demonstrated high sensing properties and a good capability to properly monitor crack evolution [28–30]. The dispersion procedure has also been optimized in order to achieve a degree of good homogenization without any substantial detriment to the mechanical properties [31]. This is achieved by means of ultrasonication of a CNT dispersion in an aqueous solution, which is assisted by the addition of a surfactant, namely, sodium dodecyl sulfate (SDS). The addition of SDS improves the mechanical dispersion of the CNTs in the aqueous solution [32–35].

The effect of CNT dispersion on the mechanical and electrical properties of carbon fiber reinforced plastic (CFRP) bonded joints in their initial state has been characterized in previous works [31]. It has been concluded that these CNT adhesive films do not induce a detrimental effect on mechanical performance and they have proved to have excellent monitoring capabilities by means of electrical measurements [28]. This work takes a further step by analyzing the potential and applicability of these proposed bonded joints under aging conditions.

The amphiphilic behavior of SDS [36,37] plays an important role in the aging properties of adhesive joints. For this reason, immersion tests have been carried out in CNT-doped adhesive films once cured by varying the amount of SDS. In addition to this, single lap shear (SLS) tests have also been conducted in standard coupons in order to see the effect of water and temperature aging. The main application of these reinforced joints is the SHM. The electrical response has been also monitored during these tests so that the electrical properties can be better characterized in order to obtain a deeper knowledge of the aging mechanisms.

## 2. Materials and Methods

### 2.1. Materials

The multi-wall CNTs used for this study were NC7000 supplied by Nanocyl, with an average diameter of 10 nm and a length of up to 2 μm.

The adhesive was a FM300K adhesive film, supplied by Cytec. This is an epoxy-based adhesive with a knit tricot carrier, which allows enhanced bondline thickness control. It has a high elongation and toughness, together with an ultimate shear strength of 36.8 MPa. It is suitable for bonding metal-to-metal and CFRP-to-CFRP systems.

CNT dispersion takes place by means of ultrasonication by using a previously optimized dispersion procedure [31]. It consists of a 20-min ultrasonication of a CNT aqueous solution at 0.1 wt%.

The disaggregation of larger agglomerates is enhanced by the addition of a SDS surfactant. To study the influence of this surfactant on the aging properties, the amount of SDS was fixed at 0.00, 0.25 wt% and 1.00 wt%.

After the dispersion procedure, the CNT suspension is sprayed over the adhesive surface prior to curing at a pressure of 1 bar at 40 cm for 0.5 s in order to achieve good homogenization of the CNTs over the surface.

In order to see the effects of aging, two types of specimens were prepared. One was the adhesive without substrate once cured—named in-bulk adhesive—in order to see the water uptake without any influence of the CFRP substrates. The second was the SLS specimens, which were made by secondary bonding of unidirectional CFRP substrates. The curing cycle was set for both the cured adhesive and the SLS joints in a hot press, as shown in Table 1. To improve the interfacial adhesion, the substrate surfaces were brushed.

**Table 1.** Cure-cycle parameters of secondary bonding.

| Parameter | First Stage | Second Stage |
|---|---|---|
| Pressure | Ramp from 0 to 0.6 MPa for 15 min | 0.6 MPa for 90 min |
| Temperature | Ramp from 25 to 175 °C for 45 min | 175 °C for 60 min |

### 2.2. Aging Tests

Cured adhesive and SLS specimens were subjected to aging conditions by immersion in distilled water at 60 °C, similar to some found in the literature [38]. Prior to immersion, the samples were dried in an oven at 50 °C for three days until weight loss was not observed between one and the next measurement. The aging time was set at 2 weeks (14 days) for the in-bulk adhesive, and up to 2 months (60 days) for SLS specimens. The reasons for these different immersion times were due to differences in the nature of each material and the exposed area subjected to water uptake. Adhesives tend to reach the water uptake saturation before the composite substrates [39] and in the case of the in-bulk adhesive, the exposed area of the adhesive is higher than in the CFRP joints. Therefore, the process of water uptake is accelerated [40].

Water absorption was measured in the in-bulk samples in their initial state and 1, 2, 3, 4, 7, 10 and 14 days after immersion. The water uptake was calculated by comparing the measured weight after immersion and the initial one in which is supposed that the samples were totally dry.

### 2.3. Electromechanical Tests

As commented before, bonded joints were subjected to SLS tests in order to study the aging effect on the electromechanical properties. They were conducted in three specimens for each condition (neat adhesive without CNTs and with 0.1 wt% CNTs with 0 wt%, 0.25 wt% and 1 wt% SDS). The tests were made according to standard ASTM D 5868-95 issue 01 using substrates of $100 \times 25.4 \times 2.5$ mm with an overlapping area of $25.4 \times 25.4$ mm at a test rate of 13 mm/min. They were performed in a universal tensile Zwick machine.

Simultaneously, the electrical response was also monitored. Electrodes were made of copper wire sealed with silver ink in order to ensure a good electrical contact with the substrate surface. To protect the electrodes from environmental influences during testing, an adhesive layer was used. The measurements were carried out by an Agilent 34401A hardware and they were correlated to the mechanical response given by the tensile machine.

### 2.4. Characterization

Differential scanning calorimetry (DSC) measurements were conducted in a Metter Toledo mod 821 apparatus for the in-bulk adhesive. Two scans were carried out according to the standard ISO 11357-2:13, at 10 °C/min from ambient temperature to 250 °C. The glass transition temperature ($T_g$)

was determined as the turning point of the heat capacity change. Two specimens of the non-aged- and 14-day-aged in-bulk specimens were measured in order to see the water uptake effect in the physical properties of the neat and CNT-doped adhesive.

## 3. Results

This section presents an analysis of the physical and mechanical evolution of SLS joints under aging conditions. First of all, the water uptake measurements for the in-bulk specimens are shown. Then, the mechanical properties of the SLS joints are discussed and finally, their electromechanical behavior is characterized.

### 3.1. Water uptake Measurements

Figure 1 shows the water uptake in terms of percentage of the initial weight for the in-bulk cured samples at each condition. The graph is in good agreement with the typical behavior of water uptake for this kind of samples, previously stated in other studies [38,41,42]. It is observed that water uptake is more prevalent in the initial stages and then the weight gain is going less significant until the water saturation is reached at 2 weeks of aging.

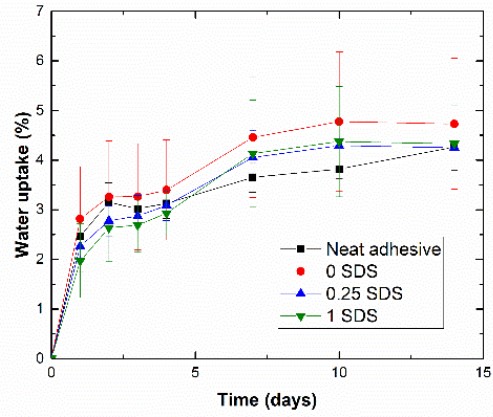

**Figure 1.** Water uptake graph for in-bulk specimens.

A similar water uptake behavior is found at every condition although some slight differences can be noticed. In this context, two opposite effects can play an important role. The first one is the hydrophobic behavior of carbon nanofillers, which can be introduced into the free volume of the polymer improving the barrier properties and leading to a reduction in the water uptake [43,44]. The second one is the amphiphilic behavior of the SDS that remains attached to the CNT surface [45,46], which can lead to an increase of the water absorption induced by the hydrophilic head groups [47]. In addition, CNT dispersion also plays a significant role. A poor dispersion can induce the presence of larger agglomerates, higher heterogeneity and higher distributed porosity. This irregular distribution leads, thus, to an irregular effect of the barrier properties of CNTs, which, in combination with the higher porosity, promotes a higher water uptake. However, a better dispersion of nanofillers improves the barrier properties leading, thus, to a lower water uptake.

The combination of these effects, as shown in the schematics of Figure 2, thus, explains the slight differences observed for each condition. In the case of the CNT reinforced adhesive without surfactant, a poor CNT dispersion is achieved, as stated in previous studies [31], so that the hydrophobic effect of CNTs is not so prevalent. Alternatively, the samples with 0.25 wt% and 1 wt% SDS show a similar trend, with a slightly lower water uptake than the sample without surfactant. In this case, the effect of the better CNT dispersion achieved was slightly prevalent over the amphiphilic effect of SDS surfactant. In the case of the neat adhesive, the water uptake was given directly by the physical behavior of the epoxy matrix.

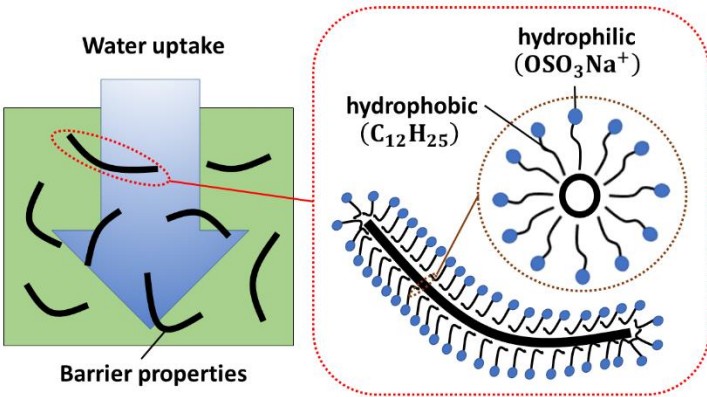

**Figure 2.** Schematics of combined effect of the water uptake (**left**) and amphiphilic behavior of sodium dodecyl sulfate (SDS) (**right**).

*3.2. DSC Measurements*

Table 2 shows the $T_g$ values at different testing conditions for the in-bulk specimens. In the initial state, a drastic reduction of $T_g$ was observed when comparing the non-doped with the doped adhesive. This means that CNTs accelerated the curing process, leading to the maximum conversion point of the system ($T_g \sim 150$ °C). This affirmation was supported by the measurements of the second scanning, where the $T_g$ of all the samples were close to 150 °C, indicating the point of the maximum conversion of the resin. In addition, by observing the $T_g$ of the aged samples, a significant decrease in comparison to the non-aged specimens was observed when adding CNTs, resulting in a similar glass transition temperature than for the neat adhesive. However, by observing the $T_g$ obtained in a second scanning, it reaches the point of the maximum conversion in every case. This indicates that there was a plasticization effect caused by the water absorption in the case of CNT-doped samples. In the case of the neat adhesive, no significant differences were found when comparing aged and non-aged samples, so the plasticization effect was similar for the non-aged and aged samples. This affirmation was given by the fact that the network of the neat adhesive was not initially totally cured, with the plasticization effect less prevalent due to water absorption. It is important to note that the selected curing cycle was the same as that given by the supplier.

**Table 2.** Glass transition temperature for different in-bulk conditions.

| Condition | Non-Aged $T_g$ (°C) | | 2-Week-Aged $T_g$ (°C) | |
|---|---|---|---|---|
| | 1st Scanning | 2nd Scanning | 1st Scanning | 2nd Scanning |
| Neat adhesive | 117.0 | 146.0 | 115.5 | 146.0 |
| 0.00 SDS | 144.0 | 148.0 | 119.0 | 146.0 |
| 0.25 SDS | 148.5 | 149.5 | 118.0 | 148.5 |
| 1.00 SDS | 146.0 | 147.5 | 109.0 | 149.5 |

Alternatively, when comparing the CNT-doped samples, it was observed that the addition of surfactant results in a more drastic reduction of the $T_g$, implying, thus, a higher plasticization effect. In order to better explain the possible effects that can take place in the material, it was necessary to focus on the mechanical testing of the SLS joints.

*3.3. Single Lap Shear Tests*

Figure 3 shows the lap shear strength (LSS) of the SLS specimens for each condition. It was observed that the LSS strength was significantly affected by the aging conditions. In every condition, a significant decrease of the LSS was observed after 1 month of aging while, for most of the cases, a slight recovery of the LSS was noticed by increasing the aging time. This different behavior can be

explained by attending the water uptake results and also by the different chemical interactions inside the adhesive joint.

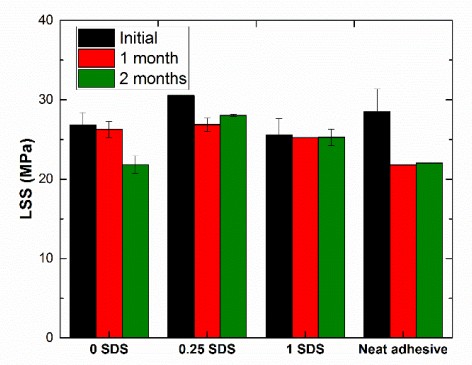

**Figure 3.** LSS of adhesive joints for each testing condition.

First of all, it is necessary to understand the role of aging mechanisms due to water absorption. The water uptake generally induces the creation of weak hydrogen bonds, leading to a swelling of the polymer chains and causing, thus, a drastic reduction of the physical properties. This effect is called plasticization and is detrimental to the mechanical properties [41]. However, after this initial stage, a slight recovery is generally observed. This has already been shown in other studies [43,48,49] and can be explained by a change in the mechanism of water absorption. After reaching the water uptake saturation, longer aging times can induce a transformation of the weak hydrogen bonds into multiple chemical connections between the water molecules and the polymer chains, promoting an increase in secondary crosslinking, thus leading to a stiffening of the material and also an embrittlement, as observed in the examples given in Figure 4a. This cannot be confirmed by the $T_g$ as there was only one measurement. Therefore, it is necessary to focus on the mechanical response of the adhesive joints, particularly on the displacement at failure.

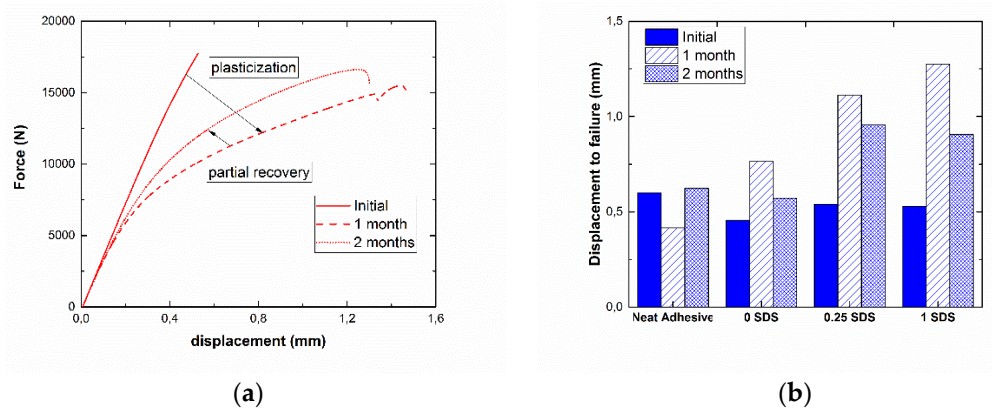

(**a**)                    (**b**)

**Figure 4.** Mechanical effect of water uptake showing (**a**) a plasticization stage and (**b**) the displacement at failure.

In the particular case of the neat adhesive, there was only a partial conversion of the epoxy matrix at the initial state, as the $T_g$ was significantly below the maximum value (~150 °C). Therefore, the water uptake does not affect significantly the plastic properties of the epoxy matrix, as it presents a more prevalent plasticization effect at the initial state. This was confirmed by a slight variation of the displacement at failure with aging time, as shown in Figure 4b.

However, in the case of CNT-doped adhesive joints, there are some different mechanisms. Here, the CNT addition itself and the dispersion state, dominated by the surfactant addition, play an important role in the aging mechanisms. As commented previously, a good dispersion implies a

high homogeneity of CNT distribution in the matrix, improving the barrier properties. This acts in an opposite way to the amphiphilic behavior of SDS. In addition, the SDS also induces variations in the chemical interactions between the epoxy matrix and the CNTs, leading to a more drastic reduction of the $T_g$, as shown in Table 2. This was also observed by the displacement at failure shown in Figure 4b. In this case, the higher the SDS content, the higher the displacement at failure was. Moreover, it can be noted that the displacement at failure decreases after 2 months of aging. This was explained by the stiffening effect induced by the secondary crosslinking, which also leads to an embrittlement of the material, as previously stated. Therefore, the combination of the two effects explains the initial reduction of the LSS after 1 month and the general recovery for longer aging times.

In the case of the CNT sample without surfactant, a slight reduction of the LSS was observed after 1 month (~2%), while the effect on the mechanical properties was much more prevalent after 2 months, leading to a LSS reduction of more than 18%. In this case, the role of CNT dispersion was even more critical, as the absence of any surfactant leads to a much lower homogeneity of CNT distribution, inducing some areas with very high CNT content, acting as stress concentrators. Here, the embrittlement effect was more prevalent than the stiffening due to secondary crosslinking. This was confirmed by a higher relative reduction in the displacement at failure after 2 months of aging, similar to that initially obtained.

The higher displacement at failure observed in aging conditions for the samples with a higher amount of SDS can be explained accordingly to the presence of polar heads. They could interact with the matrix and the plasticizing effect is explained due to the unwinding of the chains around the nanotubes, as reported in other studies [50,51].

### 3.4. Electrical Monitoring

The aforementioned results give an initial idea of how aging conditions can affect the mechanical properties of CNT-doped adhesive film joints. In order to have a deeper understanding of aging effects on CNT-doped adhesive joints, it was necessary to analyze their electromechanical behavior. Figure 5 shows an example of electrical monitoring of a SLS specimen for different aging times. In every case, it was observed that the electrical resistance increases with displacement. This increase follows an approximately exponential behavior until failure, with the changes being more prevalent in the last stages of SLS testing. As stated in a previous study proving the monitoring capabilities of these CNT reinforced joints [28], the changes in the electrical resistance are due to the combination of two effects. The first effect was the increase of the tunneling distance between adjacent particles due to strain, leading to an increase of the tunneling resistance [52,53]. The second was the sudden crack propagation in the last stages of the tests, causing a prevalent breakage of electrical pathways through the joint. However, some important differences between the aged and non-aged specimens can be found.

By deeply analyzing the curves for the sample with 1 wt% SDS, the electrical behavior as a function of the applied strain changes was observed from the initial state to the 1-month-aged sample. In the aged sample, softer behavior was observed, due to the plasticization effect. This effect causes a steadier response of the material, with no abrupt changes in the electrical monitoring, as adhesive deformation and crack propagation take place in a softer way. By increasing the aging time, as discussed before, a secondary crosslinking was induced so the effect of plasticization was reduced, showing a mixed behavior between the initial state and the 1-month-aged sample, as noticed in the right graph of Figure 5c, where an abrupt change of the electrical behavior was observed.

The sample without surfactant shows similar behavior. At the initial state, some abrupt changes in the electrical resistance were observed, while the effect of plasticization was clearly shown after 1 month of aging (left graph of Figure 5b). By increasing the aging time, the stiffening effect of the secondary crosslinking was also observed by abrupt changes in the electrical behavior. In this case, as noticed before in the mechanical response, the behavior of the 2-month-aged sample was more similar to the non-aged one.

These initial results can give a good qualitative approximation of how aged and non-aged samples behave and prove the capability of CNT reinforced joints to properly monitor their mechanical behavior by means of electrical measurements. However, from the electrical response, it was possible to obtain estimations regarding damage evolution. To achieve this purpose, a simple analytical model, based on the tunneling effect of CNT reinforced polymers is proposed.

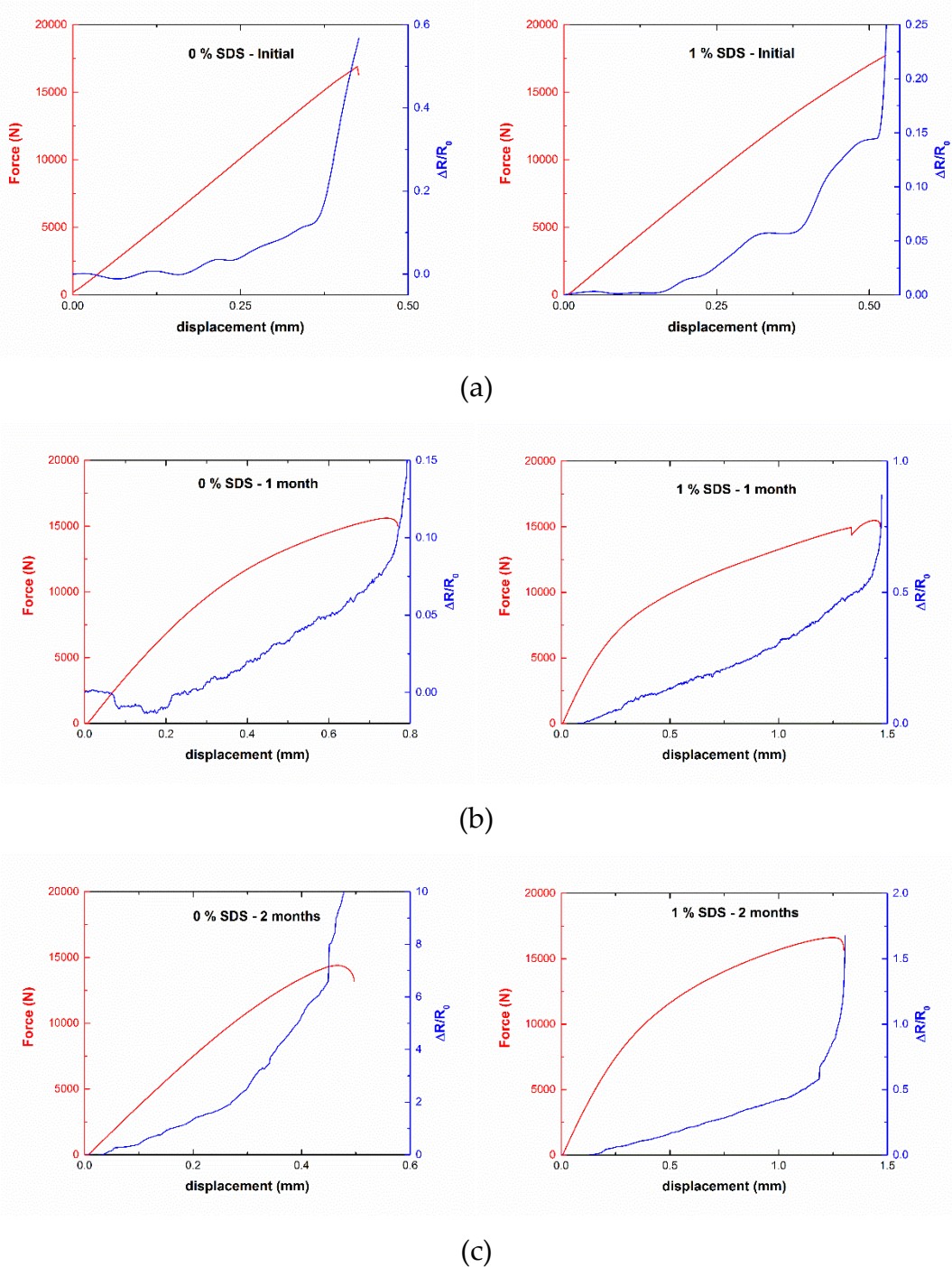

**Figure 5.** Electromechanical curves for single lap shear (SLS) specimens with (**left**) no surfactant and (**right**) 1 wt% SDS at (**a**) initial state and (**b**) after 1 and (**c**) 2 months of water immersion.

### 3.5. Theoretical Approach

The CNT-reinforced adhesive film was modeled as a nanocomposite with an homogeneous CNT distribution. The electrical mechanisms inside a nanoparticle network are given by the intrinsic electrical resistance of the CNTs themselves, the contact resistance between nanotubes and the tunneling resistance between near CNTs. In this particular case, the variation of electrical resistance due to applied strain are mainly dominated by the tunneling effect as intrinsic and contact resistance are assumed to be invariable with applied strain as stated in other studies [54].

Therefore, the changes in the electrical resistance can be divided into two terms, the first one, correlated to the variations due to the applied strain, which depends on the changes due to the tunneling effect between adjacent nanoparticles and the second one, which is correlated to the breakage of electrical pathways due to the effect of damage accumulation, as shown in the schematics of Figure 6, leading to the following expression:

$$\Delta R_{total} = \Delta R_{tunnel} + \Delta R_{damage} \tag{1}$$

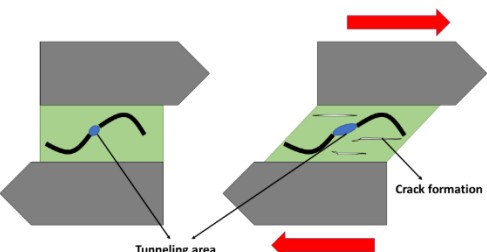

**Figure 6.** Schematics of electromechanical behavior in a SLS test showing the increase of tunneling distance and crack nucleation inside the adhesive.

The tunneling effect can be calculated by using the well-known Simmons formula for the tunneling resistance [55]. It has been proved to be an effective way to predict the electromechanical response of a strained CNT network as it is the most dominant electrical transport mechanism. It takes several aspects such as the polymer type and the contact area between adjacent CNTs:

$$R_{tunnel} = At \cdot e^{bt} \tag{2}$$

where $A$ and $b$ are two constants depending on the CNT geometry and matrix barrier characteristics and $t$ is the tunneling distance, which changes with the applied strain.

The changes due to damage accumulation are not easy to model. There are many studies investigating this effect by proposing different damage evolution laws [56,57], but the particularities of the tested systems make the damage calculation very difficult. Therefore, damage accumulation is estimated by comparing the measured changes in the electrical resistance and the known tunneling effect.

$$\Delta R_{damage} = \Delta R_{measured} - \Delta R_{tunnel} \tag{3}$$

For this purpose, the initial tunneling distance is calculated as the distance that best fits the initial changes of the electrical resistance, where no damage is supposed.

Figure 7 shows the comparison between the theoretical line, taking only the tunneling effect and the experimental measurements for aged and non-aged samples with 1 wt% SDS into account. The pattern areas indicate the differences between the theoretical and the experimental ones being, thus, the damage accumulation during the SLS test.

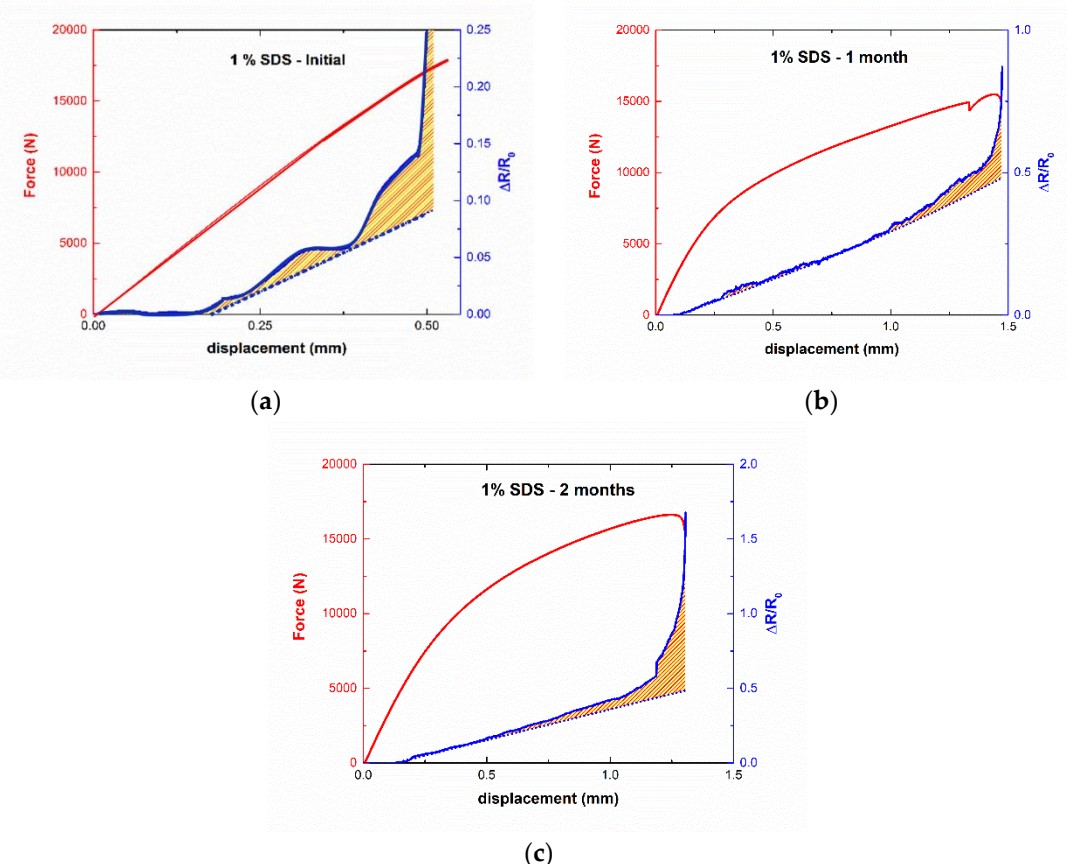

**Figure 7.** Damage accumulation evolution (red pattern area) by comparing experimental measurements (solid lines) and theoretical predictions (dashed lines) for 1 wt% SDS samples at (**a**) initial state and (**b**) after 1 and (**c**) 2 months of aging.

The non-aged sample shows significant irregular behavior. The threshold for damage accumulation was observed at ~0.25 mm displacement. After this point, the evolution of damage accumulation was very irregular. This can be explained due to the brittle mechanisms dominating the mechanical behavior of the adhesive. Secondary cracks start to nucleate and then they coalesce around the main crack [58], inducing a higher breakage of electrical paths in a similar way than that observed in other studies for fatigue testing [59]. This nucleation was not uniform so the unstable damage accumulation was explained. In the case of 1-month-aged samples, this damage accumulation starts to take place at 1 mm of displacement, that is, much later than in the non-aged specimen. This is in good agreement with the stated conclusions from water uptake and LSS measurements, as the induced plasticity in the first stages of water uptake avoids the early crack nucleation inside the adhesive joint. After that, damage accumulation takes places in a sudden way, that is, the cracks start to nucleate and then immediately coalesce. The 2-month-aged sample has a damage threshold of 0.5 mm, lower than in the 1-month-aged sample, due to the stiffening effect of the change in the water absorption mechanisms discussed above. Then, a softer evolution of damage accumulation is observed and finally, in the last stages of SLS tests a rapid coalescence takes place until final failure.

The previously described behavior was similar in the case of the 0.25 wt% SDS samples. However, in the case of the samples without surfactant, the electromechanical behavior shows some slight differences regarding the 1 wt% SDS samples, especially, concerning the sensitivity of the electrical response. Figure 8 presents the comparison between the theoretical and the experimental lines at different aging times. At the initial state, similar behavior for the 1 wt% SDS samples was observed with abrupt changes in the electrical resistance, inducing a high damage accumulation rate due to the rapid nucleation and coalescence of micro-cracks inside the material. The 1-month-aged specimen

shows a much softer behavior, as expected due to the plasticization effect of the water uptake process. The threshold of damage accumulation was observed at a 0.7 mm displacement, that is, much later than for non-aged samples. However, unexpected behavior was observed for the 2 month-aged sample. In this case, the threshold for damage accumulation was observed nearly at the beginning of the SLS tests, that is, earlier than in the case of non-aged specimens. In addition, the damage accumulation was very high also in comparison to the other specimens as they show a much higher sensitivity.

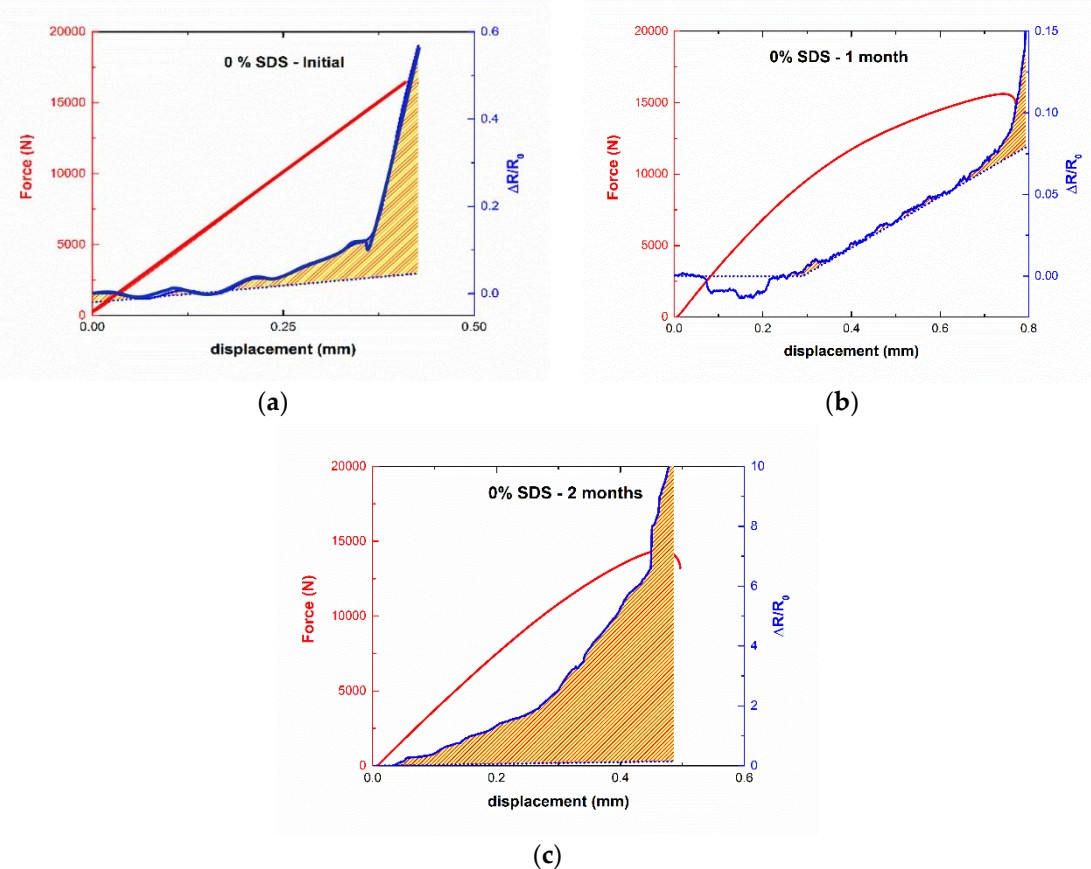

**Figure 8.** Damage accumulation evolution (red pattern area) by comparing experimental measurements (solid lines) and theoretical predictions (dashed lines) for non-surfactant samples at (**a**) initial state and (**b**) after 1 and (**c**) 2 months of aging.

This behavior can be explained by the interaction of two effects. One of them was due to the fact that the highly heterogeneous CNT distribution can induce high stress concentrations, leading to highly weak points in the matrix. This effect was also present in the non-aged and aged specimens, but in the case of the 2-month-aged one, the stiffening effect induced by the change in the mechanism of water absorption previously described can lead also to much higher embrittlement of the adhesive. This can also be stated in the reduction of the displacement at failure, so that the nucleation and coalescence of microcavities takes place more rapidly. The second one was correlated to the different interactions between the larger agglomerates of CNTs (which were much more prevalent in the non-surfactant samples) and the water molecules. This promotes a different electromechanical behavior than for the other samples.

## 4. Conclusions

SLS joints with CNT-doped adhesive films have been tested under aging conditions while their electromechanical properties have been monitored.

The water uptake measurements for the cured adhesive without substrate show that the behavior does not change significantly with the addition of both CNTs and SDS surfactant. This is explained by the combined effect of the amphiphilic behavior of the SDS and the barrier properties of CNT dispersion, acting in an opposite way.

The LSS of the bonded joints shows a general decrease after 1 month of aging because of the plasticizer effect of the water, which promotes the creation of weak hydrogen bonds. This statement was also confirmed by an increase of the displacement at failure. After 2 months of aging, there was a slight increase of LSS and a general reduction on the displacement at failure, which was explained by the secondary crosslinking that takes place due to water uptake saturation. In the case of the sample without surfactant, this behavior was slightly different because of the poor CNT distribution that can induce higher embrittlement, leading to a sudden decrease of LSS even after 2 months of aging.

Finally, the analysis of the electromechanical behavior of SLS joints confirms the previously described statements. A higher plasticization was observed for 1-month-aged specimens, while a partial recovery of the stiffness was observed after 2 months. By comparing the measured electrical response with simple theoretical calculations, it is possible to obtain the first quantitative idea of damage accumulation and how aging conditions affect the damage evolution. Therefore, this first estimation can be used to better understand the physical mechanisms taking place on CNT-doped adhesive joints under aging conditions subjected to SLS tests.

As a future work, however, it would be necessary to refine the theoretical predictions by taking some effects such as CNT distribution and orientation or the barrier properties of the epoxy matrix into account. This could give more accurate knowledge of the physical behavior of CNT-doped bonded joints under aging conditions.

**Author Contributions:** X.F.S.-R. conceptualization, methodology, formal analysis, writing—original draft preparation; A.J.-S. conceptualization, writing—review; S.G.P. supervision, formal analysis, writing—review; M.S. conceptualization; A.G. and A.U. funding acquisition. All authors have read and agreed to the published version of the manuscript.

**Funding:** This research was funded by the Ministerio de Economía y Competitividad of Spanish Government (Project MAT2016-78825-C2-1-R) and Comunidad de Madrid regional government [PROJECT ADITIMAT-CM (S2018/NMT-4411)].

**Conflicts of Interest:** The authors declare no conflict of interest.

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
