# Peer review of "Electrical Monitoring as a Novel Route to Understanding the Aging Mechanisms of Carbon Nanotube-Doped Adhesive Film Joints"

_applsci, doi:10.3390/app10072566_

Round 1
Reviewer 1 Report
This paper describes ageing mechanism for CNT-doped plastic film. I think that this paper includes neither an outstanding results nor special attraction in scientific sense, but the experiment itself was fully conducted and discussed. In addition, they mentioned that the tunneling effect in terms of Simmons law, but there are no comparative discussion. I consider that they have to discuss more about its quantitative aspect: for example, it is appropriate or not.
Reviewer 2 Report
The article is in line with the topic of the Journal and presents interesting ideas on the topic discussed. Some small revisions are needed to clarify some points:
- As shown in figure 4a the dispersion of the nanotubes plays a fundamental role in the mechanical properties of the composite. The nanotubes, when inserted inside the polymer, usually embrittle the structure, making it less tenacious due to an impediment of sliding of the chains. When the dispersant (sds) is inserted, however, there is a greater plasticizing effect (figure 4b). Have you considered if this effect could be due to an interaction between dispersant (when put in large quantities) and polymer? SDS as shown in figure 2 had polar heads and could interact with the matrix and the plasticizing effect noted could be due to the unwinding of the chains around the nanotubes. Some works shown this effect in other polymers obviously using different solvents[1,2].
- In the electromechanical tests it was evaluated the effect of the presence of water inside the structure has been evaluated as explained previously in chapter 3.3.
Some minor typos:
Line 119: Electromechanical and not electromeechanical
Figure 1: I suggest to use white instead “blanco” in the figure
I would suggest if it was possible for figures 7 and 8 to use a greater thickness for the lines drawn in order to make them clearer.
References
- Lavagna, L.; Massella, D.; Pantano, M.F.; Bosia, F.; Pugno, N.M.; Pavese, M. Grafting carbon nanotubes onto carbon fibres doubles their effective strength and the toughness of the composite. Compos. Sci. Technol. 2018, 166, 140–149.
- Paul, D.R.; Robeson, L.M. Polymer nanotechnology: Nanocomposites. Polymer 2008, 49, 3187–3204.
Round 2
Reviewer 2 Report
The authors have responded to all the points raised and I recommend the publication of the article in this form.